# Aligning the Epidemiology of Malnutrition with Food Fortification: Grasp Versus Reach

**DOI:** 10.3390/nu15092021

**Published:** 2023-04-22

**Authors:** Megan W. Bourassa, Reed Atkin, Jonathan Gorstein, Saskia Osendarp

**Affiliations:** 1Micronutrient Forum, Washington, DC 20005, USA; 2Bill & Melinda Gates Foundation, Seattle, WA 98109, USA

**Keywords:** fortification program design, nutrient adequacy, epidemiology, large-scale food fortification, micronutrients

## Abstract

Large-scale food fortification (LSFF) has been recognized as one of the most cost-effective interventions to improve the intake of vitamins and minerals and decrease the burden of micronutrient deficiency. Indeed, the simple addition of micronutrients to staple foods, such as wheat, maize and rice, or condiments, including salt and bouillon, has tremendous potential to impact malnutrition. However, most LSFF programs have been poorly designed and have not taken into consideration critical inputs, including current levels of nutrient inadequacy and per capita consumption of different food vehicles when deciding which nutrients to add and at what concentrations. LSFF programs, like some other nutrition interventions, also tend to have low coverage and reach and lack monitoring to measure this and course correct. These program design flaws have resulted in limited effectiveness and have made it difficult to determine how best to harmonize LSFF with other interventions to reduce micronutrient deficiencies, including efforts to enhance dietary diversity, biofortification and supplementation. Furthermore, LSFF has often been touted as a population-based intervention, but in fact has heterogenous effects among sub-groups, particularly those with limited access to or inability to afford fortified foods, as well as those with higher physiological requirements, such as pregnant and lactating women. This article focuses on these limitations and the concerted efforts underway to improve the collection, analysis, and use of data to better plan LSFF programs, track implementation, and monitor coverage and impact. This includes a more sophisticated secondary analysis of existing data, innovations to increase the frequency of primary data collection and programmatically relevant visualizations of data of sub-national estimates. These improvements will enable better use of data to target resources and programmatic efforts to reach those who stand to benefit most from fortification.

## 1. Introduction

Approximately 40% (3.1 billion people) of the world’s population is unable to afford a healthy diet [1]. A lack of access to healthy diets puts many of the most vulnerable populations at risk of micronutrient deficiency, which can cause an increased risk of disease and mortality [2] and new data showed high levels of deficiencies with half of the world’s young children and two-third of its women begin micronutrient deficient [3]. Unfortunately, micronutrient deficiencies often co-occur, especially in populations with low dietary diversity and inadequate access to micronutrient-rich foods and have a disproportionate impact on women and children [4,5]. 

Rather than making progress toward the 2030 Sustainable Development Goals in ending hunger and all forms of malnutrition, recent reports show that we are moving in the wrong direction on all nutrition targets (except exclusive breastfeeding). There is no shortage of evidence that the pandemic, climate change and conflicts have exacerbated the dire state of global food insecurity [1,6]. Significant changes are needed to improve food systems and diets, increase micronutrient intakes and put these goals back on track. 

Large-scale food fortification (LSFF) is widely recognized as a cost-effective method to increase the micronutrient density within a food system [7]. Numerous staple food vehicles, including wheat, maize, and rice, as well as condiments and edible oils, have been fortified with micronutrients. Currently, the Global Fortification Data Exchange cites that 161 countries have mandatory or voluntary food fortification legislation for at least one food vehicle [8]. 

When well planned, LSFF has the potential to have a broad reach with significant improvements in health without extensive behavior change [9]. Two of the most notable LSFF success stories have been demonstrated through folic acid-fortified flours and iodized salt. The impact of folic acid fortification on neural tube defects is well documented, with reductions ranging from 15 to 58% in countries around the world [10,11]. For iodized salt, the Iodine Global Network estimates that 111 countries around the world now have adequate iodine intakes, and the prevalence of goiter is virtually eliminated [12]. Salt iodization has reduced the odds of goiter by 74%, according to a recent meta-analysis [9].

While these successes demonstrate that LSFF can have a significant impact on the health and well-being of a population, there are several factors related to design and monitoring that hold it back from reaching its full potential. Data are needed to appropriately design and monitor LSFF programs to be as effective as possible at reducing the burden of micronutrient malnutrition [13,14,15]. This article will focus on the data gaps that are hampering LSFF programs and some potential solutions that are being developed to make sure that the target populations are being reached. 

## 2. Measuring LSFF Program Coverage

Data are needed in each step in the planning and monitoring process to ensure programmatic success. Ideally, a micronutrient deficiency is identified through nationally representative biomarker data; but other indicators, such as dietary intake surveys or health outcomes, can be useful. Individual-level dietary-intake data should be consulted for nutrient intakes and consumption of potential vehicles, but few low- and middle-income countries (LMICs) have these data [16]. Secondary data sources here can also be useful in understanding the availability of a micronutrient in the food supply through food balance sheets (FBS) or household consumption and expenditure surveys (HCES), which provide household-level information on foods purchased [17]. 

Once the need for an LSFF intervention has been established, data are needed to monitor the program and assess nutrient intakes and consumption patterns of food vehicles to appropriately inform LSFF programs [18]. As a result, some LSFF programs choose vehicles that are not readily fortifiable (e.g., industrially processed) or that are not widely consumed by the target population. Thus, many programs do not have the desired impact in reaching either the entire population or those in the most vulnerable groups for micronutrient deficiency. 

The Fortification Assessment Coverage Toolkit (FACT) is a survey instrument designed to assess the coverage of a fortification program that was developed by the Global Alliance for Improved Nutrition (GAIN) [15,19]. The FACT survey examines three different aspects: the consumption of the food vehicle, the consumption of a fortifiable vehicle, and the consumption of a fortified vehicle. These surveys have now been conducted in several countries, and they have highlighted the major bottlenecks that LSFF programs face [20].

Figure 1 shows an example of the FACT survey data on maize flour in three different African locations with mandatory maize flour fortification [20]. In South Africa (Eastern Cape), nearly everyone surveyed consumes maize flour. It is almost all fortifiable, and most people are indeed consuming fortified maize flour. This is an example of a fortification program where the vehicle is widely consumed and highly fortifiable, and the population is consuming the fortified vehicle. In Tanzania, by contrast, maize flour is widely consumed, but very little of it is fortified, likely because there is a large amount of home production of maize flour [20]. A third example from Lagos, Nigeria, shows that maize flour is rarely consumed in this part of the country; even less of it is fortifiable, and nearly no one consumes fortified maize flour. As a result, in Tanzania and Lagos, few people consume the fortified maize flour either because the maize consumed is not fortifiable or the vehicle is not consumed [20]. 

In these examples, it is also important to appreciate the diversity within a country that can vary geographically, among ethnic groups, and the local availability or accessibility of foods that can dramatically influence food consumption patterns. For example, in Kano, Nigeria, a much larger proportion of the population consumes maize flour, and much of it is fortifiable (approximately 84% for both) compared with Lagos [20]. However, by considering these factors, such as these, ideally, before an LSFF program is initiated, decisions can be made on the most appropriate vehicles that have the greatest potential to reach the intended populations. 

## 3. LSFF Program Equity

While LSFF is considered a population-based approach, it often has heterogeneous effects. Thus, another way to evaluate an LSFF program is to understand whether the most vulnerable populations, who have the greatest potential to benefit, are being reached. For example, those with limited access to or inability to afford adequately fortified foods, or those who may produce their own foods, may not consume fortified foods [20,21,22]. In many places, the poorest and most vulnerable populations may make or process their own foods, such as fish sauce, milled rice or wheat flour, and are not consuming industrially processed and fortified foods. Additionally, those in rural locations may have limited access to fortified foods, which may not reach their local marketplaces. 

Several studies have assessed coverage based on geographic location (e.g., urban or rural) and socio-economic status (SES) and found large differences in consumption of fortified foods between sub-groups across food vehicles [20,23,24,25]. For example, a study of vitamin A-fortified cooking oil (palm or soybean) in Bangladesh showed that while cooking oil was prevalent, there were significant differences in coverage based on the multidimensional poverty index (MPI) and urban versus rural populations. Rural populations were less likely to have purchased fortifiable oil, as were those with a high risk of poverty (high MPI) [23]. These groups likely have the greatest need for vitamin A-fortified foods but are not being reached by the current program [23].

An analysis of salt fortification coverage has similar findings in several countries. Out of the ten countries assessed, six had significantly fewer households in rural areas with salt-containing iodine (as measured by the collection and testing of salt samples from households) [24]. Coverage differences based on SES also showed that those with lower SES were significantly less likely to have iodized salt in seven of the eight countries studied. The authors noted that only one country surveyed (Uganda) had an equitable iodized salt fortification program [24]. The success of the program in Uganda is in part due to efforts to consolidate the salt industry and strong quality control and monitoring, and enforcement (discussed in the following section) [18,24]. While such practices can occur at the detriment of smaller-scale producers, it was an effective means of achieving universal salt iodization in Uganda [26].

Given the challenges in reaching the rural and poor (urban and rural) populations, it may be necessary for some settings to consider alternative distribution mechanisms to ensure that the most vulnerable groups are reached. Using public distribution systems (PDS) could be an option for reaching these populations when a market-based approach is not having the desired impact. For example, rice distributed through PDS in Telangana, India, reaches approximately 2.1 million of the 35 million people in the area. If these households consumed fortified rice, more than 80% of the RDA for several micronutrients could be met for women and children [25]. However, the authors concluded that a large-scale market-based approach for fortified rice is not feasible here due to the high prevalence of home rice milling and a fragmented rice distribution system [25]. 

As with many public health interventions, designing an equitable LSFF program with good coverage is a challenge. Lessons can be learned from countries that have achieved greater success in designing and implementing equitable programs, such as the consolidation of fragmented industries, improved monitoring and enforcement, and distribution of fortified foods through public distribution systems [25,27]. Nevertheless, additional methods may need to be employed to reach other vulnerable populations, such as those with higher physiological requirements, including pregnant and lactating women. As fortification programs aim to limit the percentage of the population receiving an excess of the fortified micronutrient, fortification programs are often not designed to meet the high needs of pregnant and lactating women. Additionally, small children 9–24 months of age typically are not able to consume enough fortified staple foods to meet their relatively high micronutrient needs [28]. As a result, it may be necessary to include targeted fortification, supplementation, or alternative interventions to meet the needs of these groups. 

## 4. Monitoring and Evaluation of LSFF Programs

Once an LSFF program is initiated, data collection in the form of continuous monitoring is required to sustain the program and ensure compliance. Cambodia provides a useful lesson in the importance of sustained monitoring and enforcement of LSFF programs. Between 2005 and 2011, approximately 83% of households were reached with iodized salt. However, when a donor stopped providing the fortificant (potassium iodate) in 2010, there was a dramatic decline in the availability of iodized salt in urban markets by 2014. The reliance on a donor for premix limited the sustainability of the salt program in Cambodia, but recent recommitment from the government has helped revive the program. Following the data indicating the program had deteriorated dramatically, their political recommitment to monitor and enforce salt iodization. This monitoring and enforcement of the salt program, including the creation of a logo for certified iodized salt. While Cambodia has not yet regained the level of fortification of 2011, today, the program is back on track with Increased availability of iodized salt [29]. 

One of the more recent advances that could transform the data landscape is the digitization of quality assurance and quality control in food processing facilities [30]. These methods could not only monitor compliance but provide useful information about the quantities of fortified foods being produced [30]. This could potentially be linked, through quick-response (QR) codes, to data collection in marketplaces that could further aid data collection on the availability of fortified foods. However, the challenge of encouraging the private sector to share proprietary data will likely remain. 

## 5. Program Co-Coverage and Risk of Excess Intake

An additional data consideration for an LSFF program is the potential for the co-coverage of interventions and the risk of excess intake. In some LMICs, there is a lack of coordination of micronutrient interventions, which most often occurs when supplementation and fortification programs are overseen by different groups [31]. A recent report also found a lack of regulations requiring the coordination of micronutrient interventions to prevent excessive intake [31]. 

One of the most notable examples of this comes from the layering of vitamin A programs in places where dietary vitamin A augmented by supplementation and fortification has led to excess intake. In Zambia, Malawi and South Africa, vitamin A deficiency has been common among some groups of children, but the layering of interventions has put them at risk of hypervitaminosis A [32,33,34]. In these examples, many of the children recently consumed vitamin A in their diets (primarily from sheep liver or fruits and vegetables high in vitamin A), in addition to good coverage of vitamin A supplementation and fortification programs. A careful evaluation of the impact of a combination of different interventions on intakes of population subgroups is required to ensure that groups are not receiving excess micronutrients and to continue the correct targeting of programs [32,33,34].

Recent changes to Guatemala’s vitamin A interventions have been implemented to reduce the risk of excess intake [31]. Recent improvements in the fortification of sugar with vitamin A have led to women of reproductive age receiving 235% of their estimated average requirement, and there were notable declines in vitamin A deficiency in children 12–59 months of age [35]. However, other data suggested that children under two years of age in very poor regions were still at risk. As a result of these data, the policy for biannual high-dose vitamin A supplementation for children was revised to include only children 6–11 months rather than 5–59 months [36]. This is a good example of monitoring of multiple interventions to ensure that the co-coverage of programs is not putting any subgroup of the population at risk of exceeding the UL for a particular micronutrient. 

Another form of overlapping programs that could lead to excess intake is in the form of iodized salt and its presence in processed foods. This was recently proposed as an issue in Ghana, where despite mandatory iodization of salt, only about one-third of households use adequately iodized salt [37]. However, the review found that voluntary fortification of salt in processed foods is likely contributing a substantial amount of iodine in the food supply, especially in urban and peri-urban areas. There is no coordination of oversight between the universal salt iodization program for salt producers and the manufacturers of processed foods. Although this evidence is limited, it raises an important issue of the total levels of iodine consumed through household salt use and the salt in processed foods that have the potential to lead to excess iodine intake, which can have detrimental effects on thyroid function [38]. 

It’s important to harmonize LSFF with other interventions to avoid the co-coverage of programs that could lead to excessive micronutrient intakes. In these contexts of co-coverage, it may be possible to reallocate some of the resources to other nutrition interventions and demonstrate greater cost-effectiveness. 

## 6. The Role of Modeling

Modeling of nutrition interventions, such as LSFF, can support the refinement of programs to assess how their program coverage and cost-effectiveness can be improved. An example of mathematical modeling of vitamin A programs in Cameroon suggested that vitamin A fortification programs in the south and cities of Cameroon were sufficient to meet the vitamin A needs of children [39,40]. In these regions, once the fortification program effectiveness has been validated, vitamin A supplementation can be safely curtailed without impact on vitamin A sufficiency. In the northern region, however, it is more useful to continue vitamin A supplementation for children because fortification programs are less likely to reach children in this region [22,23]. 

Not only could these changes in vitamin A programs reach more children, but they could also be performed at a lower cost [39,40]. The analysis demonstrated that the costs associated with the business-as-usual scenario (under the current programming) are higher than necessary. This is due to the high cost of vitamin A supplementation in much of the country. Lower costs are associated with improving fortified oils with vitamin A, implementing a fortified bouillon cube program and only supplementing children in the northern region, who are unlikely to be reached by fortified foods. To perform these analyses, detailed dietary intake data and data on program costs, in addition to subnational status data, were critical to informing the modeling estimates [39]. While adoption of these proposed modifications has been slow due to factors like engaging new political stakeholders, there are several changes that have increased the demand for data to make informed decisions and strengthened the monitoring of LSFF programs [41].

Primary data from micronutrient biomarkers or dietary intake surveys are generally preferred for mathematical modeling and were critical to the Cameroon example described above. However, some recent analyses have shown that secondary data from Household Consumption and Expenditure Surveys (HCES) can be used to inform programs with comparable results to analyses performed with primary data on dietary intakes [42,43]. 

In a mathematical modeling analysis in Malawi, HCES data was used along with Malawian and East African Food Composition Tables and fortification coverage estimates from sentinel market sites to understand the contribution of the existing fortification program to the micronutrient adequacy of diets [44]. This was compared with a no-fortification scenario and an improved enforcement scenario of the mandatory fortification policies for oil, sugar, and wheat flour [44]. 

Based on the low consumption of wheat flour, improving enforcement was unlikely to impact the micronutrient adequacy of diets [44]. However, improving the enforcement of fortification policies for oil and sugar, which are widely consumed, would likely improve the intake of vitamin A. These analyses also disaggregated data by socioeconomic status and urban/rural locations. In both geographic settings, the highest-income groups benefited more from fortified foods than those in lower-income groups [44]. This highlights the inequities of the fortification program and the need for additional interventions to reach the most vulnerable groups.

Modeling efforts such as these can support decision-makers in using data to weigh programmatic options and make more informed decisions. Some of the more intuitive answers, such as improving an LSFF program, may not have the desired impact and may not reach the populations equitably. Many of these modeling efforts are performed by international organizations, which often then advocate for change in local policies or programs. A recent analysis showed that these modeling analyses contributed to an enabling environment, but the impact on policy was limited unless it was embedded into a larger advocacy strategy [45]. Factors including targeted evidence generation at key moments in the policy cycle, local ownership of the process and data, and capacity building around the modeling methods promoted the use of the models [45].

## 7. Data Networks/Support

As the above examples have demonstrated, having data in the right place at the right time is essential to making informed-programmatic decisions for LSFF. Programs that are not informed by data may not reach the target populations for a variety of reasons, including the use of a vehicle that is not consumed or unaffordable to the poor or does not reach those in rural locations. In cases where coverage is good, there is a need for careful monitoring to prevent decline and monitor the possible risk of excessive intakes of micronutrients (e.g., from co-coverage of interventions). There are a number of tools available for decision-makers to design and monitor the performance of LSFF, such as the FORTIMAS methodology [13,15,46]. However, better support and more collaboration are needed to generate data to monitor LSFF programs and support the use of data for decision-making.

The Micronutrient Data Innovation Alliance (DInA) has been created to support the data ecosystem and bring stakeholders together. Working alongside other groups invested in micronutrient data, DInA will convene diverse national, regional, and global stakeholders to improve the availability, quality, accessibility, and use of data across the micronutrient value chain to support national-level decision-makers. Collectively, the community can leverage existing activities, avoid duplication, and ensure consensus action on priorities relevant for national decision-makers to close the data gap and expedite progress on policies and programs.

## 8. Conclusions

LSFF has been effectively employed to reduce the burden of micronutrient deficiencies, but it has even greater potential to reach more people with additional micronutrients. However, implementing LSFF programs without the requisite data can result in programs that do not have adequately fortified products or reach the target beneficiaries [20]. As a result, many LSFF programs are less cost-effective and have inadequate reach due to their poor design [27].

Data for planning and evaluating LSFF programs is an invaluable and empowering tool for decision-makers that can improve the reach and cost-effectiveness of programs. By providing actionable insights, data can ensure that LSFF programs are having the desired impact and make them more likely to reach the populations who stand to benefit the most from micronutrient interventions. This does not happen by default but rather through effective advocacy, technical support, and a sustained long-term vision for how data can empower LSFF programs. As a nutrition community, we need to work together to strengthen the capacity of national decision-makers to advocate, collect, analyze, and use data to inform LSFF program implementation and optimization.

## Figures and Tables

**Figure 1 nutrients-15-02021-f001:**
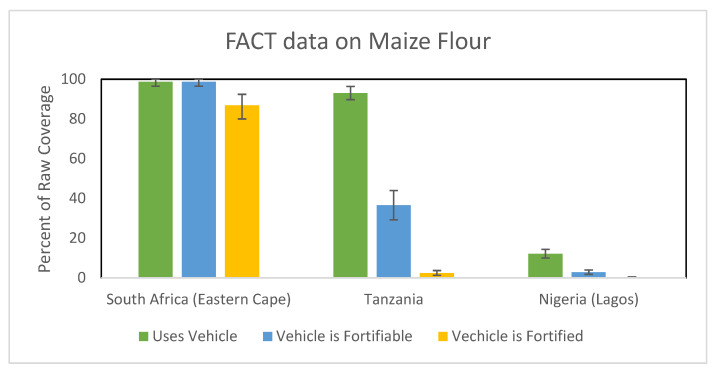
Percent of raw coverage of fortified maize flower assessed by a FACT survey in South Africa, Tanzania and Nigeria. Error bars indicate 95% CI [20].

## Data Availability

Not applicable.

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
