# Peer review of "Aligning the Epidemiology of Malnutrition with Food Fortification: Grasp Versus Reach"

_nutrients, 2023, doi:10.3390/nu15092021_

Round 1

Reviewer 1 Report

Dear Authors, excellent work done!. The presentation of data, results and discusion is brilliant.

Advertisements for candies and lollipops enriched with vitamins, iodized mineral water and cheese with calcium are common. Are such products really needed? Are they actually better than traditional food? Food fortification is the addition of one or more nutrients to a food, whether or not normally present in that food, with the aim of preventing and correcting deficiencies of one or more nutrients in entire populations or specific population groups. Other purposes of enrichment are to compensate for losses caused by processing and to increase the attractiveness of the product to the consumer. We distinguish between obligatory (obligatory) and voluntary enrichment. Obligatory fortification imposes on all producers of a specific range of foodstuffs the obligation to add vitamins or minerals in the amount regulated by relevant regulations. In some countries, it is mandatory to iodize table salt and fortify spreadable fats with vitamins A and D (with the exception of butter, which contains them naturally). Salt iodization is carried out for the prevention of hypothyroidism. Voluntary fortification leaves producers more freedom in terms of the choice of the added ingredient as well as its amount, but it is also limited by certain regulations. Only certain groups of products can be enriched with specific ingredients (e.g. vitamins B1, B2 and B6 must not be added to oils, sugar or salt, and iron and iodine - to margarines and mayonnaises). An enriching additive should not contribute to undesirable changes in color, taste or smell, change the product's functional characteristics (e.g. baking properties of flour), or shorten its shelf life. In order for the producer to be able to put information on the packaging that the product is fortified, he must add a significant amount of it - usually it is at least 15% of the recommended daily intake in 100 g, 100 ml or in a portion (if it is less than 100 g or 100 ml) . Failure to take any of these recommendations into account when fortifying food may have serious consequences. Such a case took place in the Philippines, where it was decided to fight against vitamin B2 deficiency by enriching rice. Unfortunately, this vitamin has a strong yellow-orange color, which made some rice grains significantly different from the rest. Housewives considered these grains spoiled and laboriously selected them during the preparation of meals. As a result, the venture failed. The choice of products that can be enriched is controversial. Unprocessed food (fruits, vegetables, meat, poultry and fish), as well as alcoholic beverages containing more than 1.2% vol. ethanol. However, the question remains whether the fortification of sugar, sweets or sweet breakfast cereals should be allowed, because the information on the packaging about the vitamins and minerals contained in them definitely encourages consumers to buy, and these products themselves do not contribute to the formation of proper eating habits . Food fortification undoubtedly allows for a relatively easy and socially acceptable prevention of certain nutritional deficiencies, but on the other hand it also carries a certain risk and favors its ruthless use for marketing purposes. The solution is legal supervision, monitoring and health education of the society, so that everyone can make a conscious decision to buy after getting acquainted with a properly labeled product.

I find the article extremely interesting and can be of practical use to both nutritionists and food technologists in the industry.

Author Response

We thank the reviewer for their thoughtful comments on food fortification. We agree that there are indeed many challenges that must be considered when implementing a program, but if done well there are many advantages to the population.

Reviewer 2 Report

1.      The document reads more like an opinion than a review; consider changing the manuscript type to opinion.

2.      The title and abstract are provocative.  The subsequent text would be strengthened by a greater referencing of key statements that underline the main arguments.

3.      Page 1, Lines 13-17 imply that other, non-fortification programs do not have the problems that plague fortification programs.  Is that true?

4.      Page 2, Line 61.  Does “several factors” refer to poor design and monitoring?

5.      Add references to the facts stated here:

a.      Page 2, lines 68-73, 81-86.

b.      Page 3, lines 104-106.

c.       Page 4, lines 154-59, 174-6. 

d.      Page 5, lines 211-8, 237-9. 

e.      Page 6, lines 254-6. 

6.      One of the key arguments in the manuscript bases its conclusions on an article that has incomplete data.  For example, Page 2, Lines 76-80 states "few LMICs have the requisite data to assess nutrient intakes and consumption patterns of food vehicles to appropriately inform LSFF programs [12]."   Reference 12 states "For example, in 2017 only four sub-Saharan African countries (Ethiopia, Nigeria, South Africa and Uganda) had at least one nationally representative food consumption survey (Fig. 2(a)). In contrast, a lot more countries without a national food consumption survey have introduced mandatory food fortification programmes or have the legislation mandating grain fortification (Fig. 2(b))." 

a.      In comparing Figures 2a and 2b, most of the countries in Central America and South America are coded as not having "one National Individual Food Consumption Survey by 2017".  If the authors are referring to nationally representative intake (or proxy) data, a search in the Global Health Data Exchange website suggests that several countries in this region have such information from the 2010-2017 period:

1.      Guatemala:  Guatemala National Survey of Living Conditions 2011

2.      Costa Rica:  Costa Rica National Household Survey 2016

3.      Colombia: Colombia Great Integrated Household Survey 2011

4.      Ecuador: Ecuador Living Conditions Survey 2013-2014 & Ecuador National Health and Nutrition Survey 2012

5.      Peru:  Peru National Household Survey 2016

6.      Bolivia:  Bolivia Household Survey 2016

7.      Paraguay:  Paraguay Income, Expenditures, and Living Conditions Survey 2011-2012

8.      Argentina: Encuesta Nacional de Gastos de los Hogares 2012 / 2013

9.      Uruguay:  Uruguay Continuous Household Survey 2017

b.      A search for World Bank Living Standards Measurement Study surveys may also yield household income and expenditure reports.

7.      Page 2, Line 89.  Who is “everyone” / who was surveyed?  And does it refer to the target group for fortification? 

8.      Figure 1 was not included in the manuscript submission.  

9.      Page 3 Lines 98-100.  Is it possible that maize flour is an appropriate vehicle for other states in Nigeria (i.e. other than Kano and Lagos)?  This raises another important issue:  the challenges of designing a national program when food patterns vary by geographic region (e.g. Nigeria), race/ethnic groups and other characteristics.  

10.  Page 3, lines 107-109 & 126-35.  These points are well taken for the rural poor; however what about the urban poor?  Are they not as likely to consume industrially processed foods as urban dwellers, in general?

11.  Page 3, lines 121-2.  For iodized salt, specify:  was this based on a person reporting consumption of iodized salt?  Was it based on households with iodized salt in their homes?

12.  Page 3, lines 124-5.  What can be learned from this country?  Did it have data for better decision making at the start of the program?  In other words, what can you share from this experience that supports your arguments?

13.  Page 3, lines 136-44.  This is always true—fortification is not a panacea for all micronutrient deficiencies suffered by all population groups.  Is there another lesson that can be shared?

14.  Page 4, lines 146-153.  This reflects problems with donor-driven premix purchase and not poor monitoring.  Consider changing the example. 

15.  Page 4, lines 154-9.  Add the challenge of getting the private sector to share proprietary data. 

16.  Page 4, line 168.  Sheep liver, fruits and vegetables—not sure what the point is of mentioning these.  Was the increase in these foods because of a dietary diversity program?

17.  Page 4, lines 160-79.  Consider adding experiences shared from other countries that were published in this supplement of the Annals of the NY Academy of Sciences on the risk of excessive intake of vitamins and minerals:  https://nyaspubs.onlinelibrary.wiley.com/toc/17496632/2019/1446/1

18.  Page 4, lines 181-9. Have any of these changes been made in the country?  If not, is it a good example of using modeled data for decision making?

19.  Page 4, line 195.  Practically, how feasible is supplementation targeting by region?  Is there any evidence that this has been tried (with success) in any country?

20.  Page 5, line 234.  No argument is made for why behavior change communication is needed. 

21.  Page 5, line 239.  Consider adding to the end of the sentence:  and to support the use of the data for decision making. 

22.  Page 6, line 250.  Does LSFF not now reach a large proportion of the global population, for example with salt iodization?

23.  Page 6, line 254.  What is the evidence that many LSFF do not achieve their desired impact on health outcomes?  No data on health outcomes were presented in the manuscript. 

24.  Page 6, line 261.  Is the goal to reach those who stand to benefit the most or those most in need? 

Reviewer 3 Report

A brief summary

The article has an interesting topic suitable for the selected journal, but some important changes are needed. The English language needs to be reviewed by a professional for better understanding.

General comments

Please state in the abstract what the aim of the review is.

Are the keywords well chosen? Try to add some more.

There are too few references for a review paper. It is necessary to add more recent literature.

Please follow the instructions for authors when preparing the reference list.

Make sure that all abbreviations are explained the first time they are mentioned (e.g. LMIC)

References are often not given in the text (e.g., after most paragraphs). Are these all conclusions of the author? Any statement that is not the author's conclusion based on what he or she has read must be supported by a reference. Please double check this and adjust if necessary.

Specific comments

Line 35                  What is the reference number? Please add it to the Reference list

Line 49-51            Please rewrite the sentence for better understanding

Line 75                  Please add the abbreviation

Line 83-84            Please rewrite the sentence for better understanding

Line 191-193       Please rewrite the sentence for better understanding

Line 274                References should be checked and adjusted according to Instructions for authors
